

# Bacterial communities in the digester bed and liquid effluent of a microflush composting toilet system

Laura E. Williams[1], Claire E. Kleinschmidt[1,2] and Stephen Mecca[2,†]

[1] Department of Biology, Providence College, Providence, RI, USA
[2] Department of Engineering-Physics-Systems, Providence College, Providence, RI, USA
[†] Deceased author.

Corresponding author
Laura E. Williams,
lwillia7@providence.edu

## ABSTRACT

Lack of access to clean water and sanitation is a major factor impacting public health in communities worldwide. To address this, the S-Lab at Providence College and the Global Sustainable Aid Project developed a microflush composting toilet system to isolate and treat human waste. Solid waste is composted within a filter-digester bed via an aerobic process involving microbes and invertebrates. Liquid waste may be sanitized by solar disinfection (SODIS) or slow sand filtration (SSF). Here, we used 16S rRNA amplicon sequencing of samples from a scaled-down test version of the system to better understand the bacterial component of the toilet system. Immediately after fecal matter was deposited in the test system, the bacterial community of the filter-digester bed at the site of deposition resembled that of the human gut at both the phylum and genus level, which was expected. Genus-level analysis of filter-digester bed samples collected over the next 30 days from the site of deposition showed reduced or undetectable levels of fecal-associated taxa, with the exception of *Clostridium* XI, which persisted at low abundance throughout the sampling period. Starting with the sample collected on day 4, the bacterial community of the filter-digester bed at the site of deposition was dominated by bacterial taxa commonly associated with environmental sources, reflecting a major shift in bacterial community composition. These data support the toilet system's capacity for processing solid human waste. We also analyzed how SODIS and SSF sanitization methods affected the bacterial community composition of liquid effluent collected on day 15 from the test system. Untreated and treated liquid effluent samples were dominated by Proteobacteria. At the genus level, the bacterial community of the untreated effluent included taxa commonly associated with environmental sources. In the SODIS-treated effluent, these genera increased in abundance, whereas in the SSF-treated effluent, they were greatly reduced or undetectable. By analyzing operational taxonomic units that were unclassified at the genus level, we observed that SSF appears to introduce new taxa into the treated effluent, likely from the biological film of microbes and small animals that constitutes the key element of SSF. These data will inform continued development of liquid waste handling strategies for the toilet system. Using the test system as an indicator of the performance of the full-scale version, we have shown the effectiveness of the microflush composting toilet system for containing and eliminating gut-associated bacteria, thereby improving sanitation and contributing to better public health in rural and peri-urban communities.

## INTRODUCTION

In 2015, the United Nations released a global sustainable development agenda featuring 17 goals with specific outcomes to alleviate poverty, improve public health, and strengthen environmental protection by 2030 (*UN General Assembly, 2015*). In particular, Goal 6 is focused on providing clean water and sanitation to communities around the world. Water and sanitation directly impact health and wellbeing, yet 2.1 billion people lack safely managed sources for drinking water, and 2.3 billion people lack access to basic sanitation facilities such as toilets (*World Health Organization & United Nations Children's Fund, 2017*). This disparity results in a heavy disease burden on affected communities. According to the World Health Organization, 870,000 people died in 2016 due to inadequate sanitation and unsafe drinking water (*World Health Organization, 2018*). Isolating and treating human waste is one strategy to reduce contamination of water sources and improve public health.

To this end, the S-Lab at Providence College and the Global Sustainable Aid Project (GSAP) developed a low cost, microflush toilet system that provides an effective sanitation solution for rural and peri-urban areas (*Mecca, Davis & Davis, 2013a*). The GSAP Microflush toilet is designed to use locally sourced materials so that a trained local artisan can fabricate the system. The user interface sits above a microflush valve, which uses the small volume (~150 cc) of graywater from the previous user's hand wash to move both solid and liquid waste into a filter-digester bed. The bed measures $3' \times 6'$ for the household version of the toilet and $4' \times 7'$ for a larger school block version. Liquid waste flows through a series of screens and mesh to a filtrate chamber that lies below, whereas solids are digested and composted in the layered bed by an aerobic process involving microbes and invertebrates (usually the earthworm *Eisenia fetida*). The rear cover of the GSAP toilet can be removed to retrieve the compost every 2–3 years for use in agriculture.

Liquid effluent typically exits the filtrate chamber of the GSAP Microflush toilet via a drainage pipe to a soak hole, which acts as a scaled-down version of a leach field. For communities with high water tables, there are two options under consideration for sanitization of liquid effluent: solar disinfection (SODIS) or slow sand filtration (SSF). SODIS involves collecting liquid effluent in a clear container such as a recycled water bottle and exposing the container to sunlight. This approach significantly reduces levels of bacteria, including pathogens (*Boyle et al., 2008*; *Caslake et al., 2004*). SSF treatment involves migration of liquid effluent through a naturally formed biological film (termed the schmutzdecke) containing microbes and sometimes small animals. This film, which sits atop layers of sand and gravel, removes pathogens, and other contaminants (*Elliott et al., 2008*; *Hijnen et al., 2004*).

The ability of the GSAP Microflush toilet to process waste is supported by previous studies investigating temperature distribution within the digester (*Mecca, Stifler & Beley, 2017*) and the use of SODIS for pathogen reduction in liquid effluent (*Mecca, Pellock & Bretz, 2013b*). In addition, studies of vermicomposting as a waste treatment method have
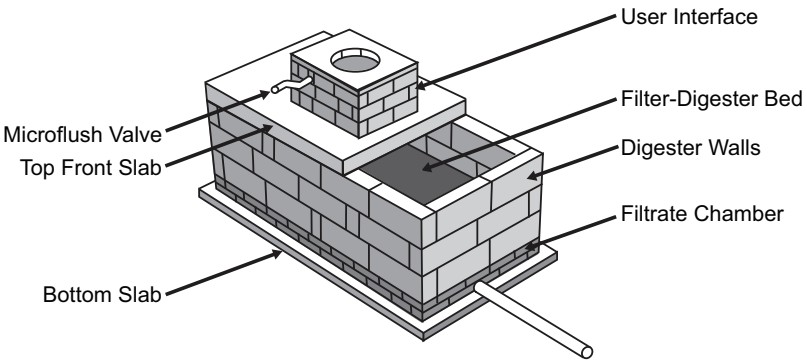

**Figure 1 Diagram of GSAP Microflush toilet test system.** When used in a community setting, the toilet is enclosed and includes a small hand wash station.   

demonstrated the effectiveness of this approach for mass reduction and pathogen removal (*Hill & Baldwin, 2012*; *Soobhany, Mohee & Garg, 2017*). However, little is known about how the bacterial community composition of the filter-digester bed within the GSAP Microflush toilet changes over time after waste is deposited or how SODIS and SSF treatment impact the bacterial community composition of liquid effluent. To investigate these questions, we analyzed samples from a test system (Fig. 1), which is a scaled down version of the field system described above. The test system had been maintained for over 3 years by adding human fecal matter and vegetable scraps multiple times per week, which simulated the use of the field system.

Using 16S rRNA amplicon sequencing, we analyzed the bacterial community composition of the filter-digester bed immediately after depositing human fecal matter, and we then sampled the filter-digester bed four times over the next 30 days to track the abundance of fecal-associated operational taxonomic units (OTUs) and assess changes in community composition. We also compared the bacterial community composition of untreated liquid effluent collected from the filtrate chamber on day 15 with samples of the same effluent treated by either SODIS or SSF to investigate the effects of these sanitization methods. These data yield insight into the bacterial component of the GSAP Microflush toilet system and will help inform further development and deployment of this valuable sanitation solution.

## MATERIALS AND METHODS

### Filter-digester bed sample collection

At time zero, human fecal matter was deposited in the filter-digester bed of the test system, then a sample of this fecal matter was taken with a sterile spoon and transferred to a sterile 50 ml conical tube. No additional materials were added to the test system for the duration of the experiment. On days 4, 10, 15, and 30, a sterile spoon was used to mix the material at the location of the initial deposit, and a sample was then transferred to a sterile 50 ml conical tube. All samples were taken at approximately the same time of day and placed at −80 °C within a few hours.

## Liquid effluent sample collection and treatment

On day 15, liquid effluent was collected from the drainage pipe connected to the filtrate chamber at the base of the test system. This 15-day period allowed time for liquid waste in the filter-digester bed to migrate through the layered bed and the series of screens and mesh separating the bed and the filtrate chamber. Samples of the collected effluent were transferred to sterile 500 ml sample bottles and stored at −80 °C as raw (untreated) liquid effluent. The remainder of the effluent was used to test two different methods for sanitizing liquid effluent. To test the SODIS method, liquid effluent was immediately poured into a non-sterile Platypus bag, which was then attached to a solar rack at a 45° angle and exposed to direct sunlight. After 24 h, a sample of the treated effluent was transferred to a sterile 500 ml sample bottle and stored at −80 °C. To test the SSF, liquid effluent was transported to the lab in five gallon closed containers and stored at ambient temperature for <24 h. To begin the filtration process, nine l of water was pushed through the filter to fill the chamber. Then, three l of liquid effluent was pushed through the filter, followed by 33 l of water. Considering the exit rate and void volume of the SSF, we calculated that the liquid effluent would exit the system after 10 h. In addition, the liquid effluent fraction was visibly different from the water fractions, which enabled us to take a sample midway through the liquid effluent fraction exiting the system. The sample was collected with a sterile 500 ml sample bottle and stored at −80 °C.

## Genomic DNA extraction

Genomic DNA was extracted from the filter-digester bed and liquid effluent samples using the PowerSoil DNA Isolation Kit (MO BIO Laboratories, Carlsbad, CA, USA). For the filter-digester bed samples, 0.26–0.28 g of sample were used. For the liquid effluent samples, samples were thawed and then centrifuged in sterile centrifuge bottles at 4,000 rpm for 15 min in two 250 ml increments to pellet bacterial cells and other solid particulates. Pellets were transferred to the PowerSoil tubes to begin the extraction process. All filter-digester bed and liquid effluent samples were processed together in one run, along with a negative kit control to test for bacterial DNA contamination in the genomic DNA extraction kit. This negative kit control was a single prep taken through all of the steps alongside the samples but with no sample added at the beginning. Genomic DNA concentration was estimated using a Qubit analyzer. The genomic DNA extract with the lowest concentration was estimated at 3.77 μg/ml. Each genomic DNA extract was diluted to this concentration using PCR grade water (MO BIO Laboratories, Carlsbad, CA, USA). Genomic DNA extracts and dilutions were stored at −20 °C.

## 16S rRNA gene amplification

Using the dilutions of genomic DNA as template, we amplified the V4 region of the 16S rRNA gene following a two-step protocol. We performed the initial amplification step with KAPA HiFi HotStart Ready Mix using primers 515F and 806R (*Caporaso et al., 2011*) modified with overhang adapters (515F_mod: TCGTCGGCAGCGTCAGATGTGT ATAAGAGACAGGTGCCAGCMGCCGCGGTAA, 806R_mod: GTCTCGTGGGCT

CGGAGATGTGTATAAGAGACAGGGACTACHVGGGTWTCTAAT). Each genomic DNA dilution was amplified in triplicate. To test for bacterial DNA contamination in PCR supplies, we included a "no template" control that used PCR grade water in place of DNA as template. For a positive control, we used genomic DNA from a mock community of 20 bacterial strains (HM277-D; BEI Resources, Manassas, VA, USA). PCR cycles were: 95 °C for 3 min, then 25 cycles of 95 °C for 30 s, 55 °C for 30 s, and 72 °C for 30 s, followed by a final step of 72 °C for 5 min. An aliquot of each reaction was analyzed by gel electrophoresis to confirm presence of a band at the desired size. Equal volumes of each of the triplicate reactions were pooled for each sample.

Pooled reactions were sent to the University of Rhode Island's Genomics and Sequencing Center, where they were cleaned using 0.7x Ampure XP beads. Cleaned reactions were quantified with a Qubit analyzer, and approximately 50 ng of DNA was used as template in the second amplification step. The only exceptions to this were the negative kit control and the no template control, which had low DNA concentration by Qubit analysis. For these, five µl of the cleaned reaction was used as template. Six cycles of amplification were performed using Phusion high-fidelity PCR Mastermix and Nextera indices. Reactions were cleaned with 0.6x Ampure XP beads, quantified with a Qubit analyzer, and diluted to 10 nM. Three µl of each dilution were pooled for the final library, along with three µl of the reactions for the negative kit control and no template control, which were less than 10 nM based on Qubit analysis. Sequencing of the library was performed on an Illumina MiSeq using a 500-cycle v2 kit.

## Data analysis

16S rRNA amplicon data was analyzed using mothur v.1.36.1 (Schloss et al., 2009), following the MiSeq SOP (https://www.mothur.org/w/index.php?title=MiSeq_SOP&oldid=9609). To assess positive and negative controls, all samples were analyzed together. Then, filter-digester bed samples were analyzed separately to investigate changes in the bacterial community of the digester over time, and liquid effluent samples were analyzed separately to compare sanitization methods. In all analyses, sequences <248 or >275 bp and sequences with ambiguous bases were discarded. Remaining sequences were aligned against Silva reference database release 123 (Yilmaz et al., 2014). Using UCHIME (Edgar et al., 2011), 2.8% of sequences from filter-digester bed samples and 3.1% of sequences from liquid effluent samples were identified as chimeric and discarded. Remaining sequences were classified using the Ribosomal Database Project 16S rRNA training set version 14 (Cole et al., 2014). For filter-digester bed samples, 42 sequences were classified as chloroplast, mitochondria, archaea, eukaryotic or unknown and removed, whereas 179 sequences were removed during this step for liquid effluent samples. Sequences were then classified into OTUs at a 0.03 cutoff, which means that sequences within an OTU are at least 97% similar to each other. Filter-digester bed samples ranged from 115,835 to 169,829 sequences, and each sample was subsampled to 115,835 sequences. Liquid effluent samples ranged from 106,154 to 164,159 sequences, and each sample was subsampled to 106,154 sequences.

### Data and code

Raw MiSeq data, mothur output files and R code used for analyses can be found at the FigShare repository: https://figshare.com/projects/Data_and_R_code_for_Microflush_Composting_Toilet_Microbiome_Project/35342.

## RESULTS

### Sequencing of mock community detected all expected taxa with a low error rate, but not all taxa classified to genus level

As a positive control, we sequenced a mock community (HM277-D; BEI Resources, Manassas, VA, USA) alongside the samples and negative controls. This mock community is comprised of known concentrations of genomic DNA from 20 different bacterial species. In the mock community sample, we detected 672 unique sequences from 181,020 total sequences. To estimate our overall error rate, we compared these sequences to the known 16S rRNA gene sequences for the species represented in the mock community. This gave an error rate of 0.006%, which suggests that data processing steps in mothur, such as removal of chimeras, were effective in reducing the error rate for the sequences.

We then clustered the sequences into OTUs using a cutoff of $\geq$97% similarity. In the mock community sample, there were 22 OTUs comprised of at least 10 sequences, and 89 OTUs comprised of <10 sequences. Of the 22 "abundant" OTUs, 18 were classified to the level of genus, whereas two were only classified to the level of Family, and two were only classified to the level of Order. Based on the expected representation in the mock community (Table S1), we matched each of the four OTUs classified to Family or Order with an expected species (e.g., Otu00046 was classified as Order Bacillales, and the mock community is expected to include *Bacillus cereus*). We tested these matches by using BLAST to align a representative sequence to the 16S rRNA GenBank database. For each of the four OTUs, BLAST analysis identified hits to the matched genus at $\geq$97% similarity. Overall, we identified OTUs corresponding to each of the 20 expected mock community species. We did not identify any unexpected OTUs comprised of >10 sequences; all of these "abundant" OTUs could be matched to one of the species included in the mock community.

Analysis of the mock community indicates that sequencing and data processing accurately identified OTUs, because all expected species were detected with a low error rate; however, it revealed limitations in OTU classification, because some mock community OTUs could not be classified to the level of genus. For filter-digester bed and liquid effluent samples, we addressed these limitations by analyzing the data at different taxonomic levels, including detailed analysis of "unclassified" OTUs.

### Negative controls show minimal contamination of materials with bacterial DNA

We performed two types of negative controls to assess bacterial DNA contamination in experimental materials. In the negative kit control to test genomic DNA extraction materials, we detected 206 unique sequences from 1,123 total sequences, which were clustered into 184 OTUs. In the no template control to test PCR materials, we detected

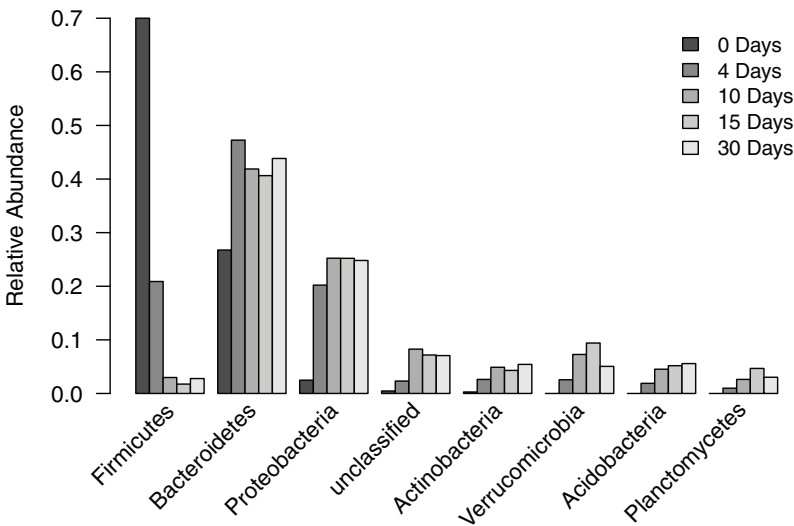

**Figure 2** Relative abundance of phylum-level groups occurring at a mean relative abundance >2% across all filter-digester bed samples.

37 unique sequences from 200 total sequences, which were clustered into 34 OTUs. By comparison, the smallest number of total sequences detected in a sample was 106,154. To further assess possible contamination of materials with bacterial DNA, we determined how many unique sequences occurred in samples and negative controls (Fig. S1). A total of 10 unique sequences were detected in samples (filter-digester bed only, liquid effluent only, or both) and both controls. A total of 10 additional unique sequences were detected in samples and the no template control, but not the kit control, whereas 116 unique sequences were detected in samples and the kit control, but not the no template control. For the kit control, some of these shared sequences may be due to cross-contamination of the kit control with sample material, because the kit control was processed alongside samples. Overall, only 1.1% of unique sequences detected in the filter-digester bed samples are also detected in controls, and only 1.6% of unique sequences detected in the liquid effluent samples are also detected in controls. This indicates that bacterial DNA contamination of materials should have minimal impact on analysis and interpretation of sample data.

## Bacterial taxa associated with deposition of fecal matter in test system are replaced over time

To understand how the composition of the bacterial community in the filter-digester bed changes over time, we collected a sample immediately after depositing human fecal matter and then after 4, 10, 15, and 30 days, during which no additional materials were added. Each sample was subsampled to 115,835 sequences. Analysis of filter-digester bed samples clustered the subsampled sequences into 3,049 OTUs, which were classified into 24 phyla and one "unclassified" group that includes all OTUs that could not be classified at the phylum level. Of these 25 groups, seven phyla and the "unclassified" group occurred at a mean relative abundance >2% across all samples (Fig. 2). None of the
remaining 17 phyla occurred at a relative abundance >2% in any of the samples, and only Chloroflexi occurred at a relative abundance >1% (in samples from day 10 to day 30).

At the level of phylum, the bacterial community of the time zero sample was dominated by Firmicutes (70.0%), followed by Bacteroidetes (26.8%), Proteobacteria (2.5%), and Actinobacteria (0.3%). By comparison, subsequent samples collected 4–30 days later showed a shift in bacterial community composition in the filter-digester bed (Fig. 2). The proportion of Firmicutes decreased to 20.9% on day 4 and further decreased to 3.0% on day 10. The proportion of Bacteroidetes increased to 47.3% on day 4, then decreased slightly but remained >40% through day 30. Proteobacteria and Actinobacteria increased in abundance from time zero to day 4 to day 10 and then remained stable. Beginning on day 4, we detected three phyla that were not present in the time zero sample: Verrucomicrobia, Acidobacteria, and Planctomycetes. These phyla were consistently detected in each of the filter-digester bed samples collected from day 4 through day 30.

To gain a finer scale perspective of the changes in bacterial community composition of the filter-digester bed, we analyzed the OTUs at the level of genus. The 3,049 OTUs were classified into 287 genera and one "unclassified" group that includes all OTUs that could not be classified at the level of genus. The unclassified group comprised the largest proportion of sequences in each sample and is discussed further below. In the time zero sample, 10 genera occurred at a relative abundance >2%, with 59.2% of sequences clustering into these groups (Fig. 3A). The genera belonged to either Bacteroidetes or Firmicutes. For each genus except *Clostridium* XI, relative abundance decreased over the subsequent samples until it was <0.02% in the day 30 sample. By contrast, the proportion of *Clostridium* XI remained above 0.8% in each sample through day 30.

In the samples collected between day 4 and day 30, new genera were detected that were not observed in the time zero sample (Fig. 3B). In particular, six genera occurred at a mean relative abundance >2% across these samples but did not occur in the time zero sample. These genera belong to Actinobacteria, Bacteroidetes, Proteobacteria, and Verrucomicrobia. This genus-level analysis suggests that the increase in abundance of Bacteroidetes, Proteobacteria, and Actinobacteria over the sampling period was due to a change in the composition of genera in the filter-digester bed rather than an increase in abundance of the same genera observed at time zero.

Because the "unclassified" group included such a large proportion of sequences in each sample, and because analysis of the mock community suggested that some "standard" taxa may not be fully classified to the genus level in our data, we analyzed unclassified OTUs on their own. Overall, this group comprised 35.6% of sequences in the time zero sample. This proportion decreased to 26.5% in the day 4 sample before increasing over the subsequent samples to 62.3%. To investigate this further, we analyzed individual unclassified OTUs. Relative abundance in the following analysis refers to the abundance of a particular unclassified OTU compared to the total number of sequences clustered into the unclassified group during the genus level analysis. In the time zero sample, six OTUs occurred at a relative abundance >2% and included 78.7% of the total sequences in the unclassified group (Fig. S2A). Each of these OTUs decreased in abundance in the subsequent samples until they occurred at <0.15% in the day 30 sample. Beginning

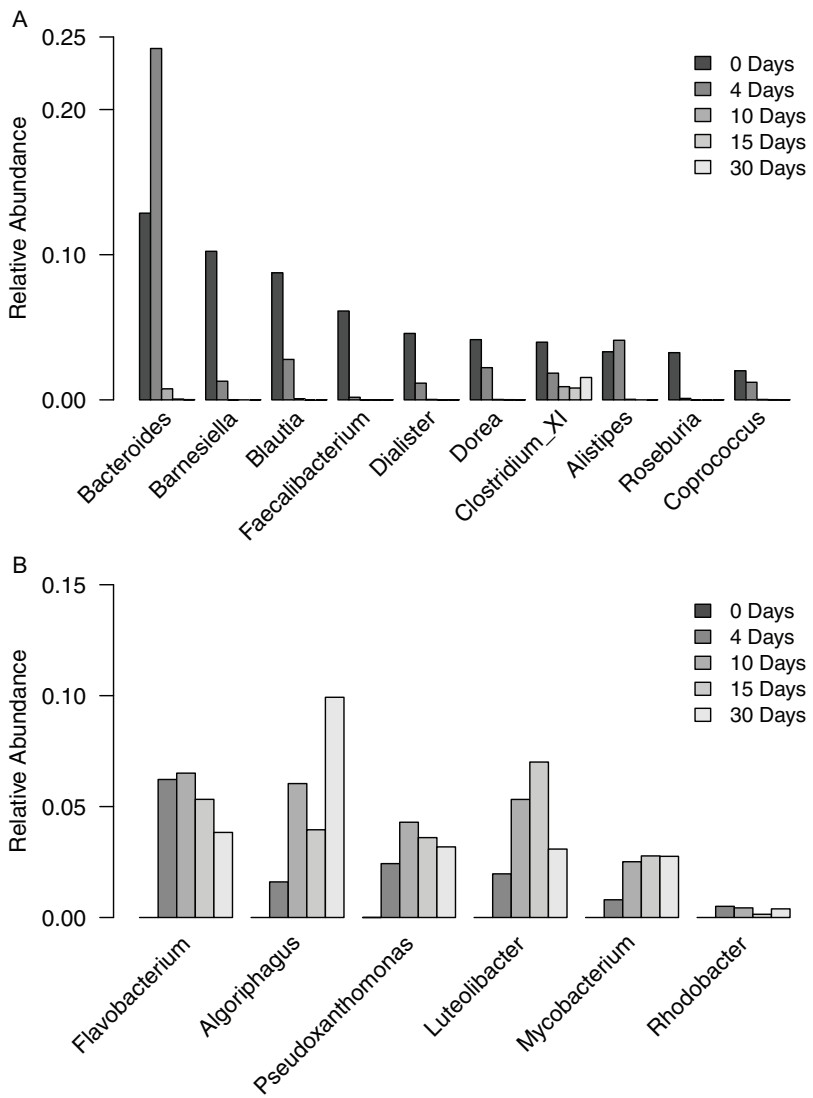

**Figure 3 Relative abundance of genera occurring in filter-digester bed samples.** (A) Genera occurring at >2% in the time zero sample, shown for all filter-digester bed samples. (B) Genera occurring at a mean relative abundance >2% in the four subsequent samples (day 4–day 30), shown for all filter-digester bed samples.

with the day 4 sample, new unclassified OTUs were detected. In particular, seven OTUs occurred at a mean relative abundance >2% across the samples collected on and after day 4 (Fig. S2B). These OTUs included 23.5–46.4% of the sequences in the unclassified group of each sample; however, none of them were detected in the time zero sample. These data parallel the genus level data and further illustrate a shift in the bacterial community composition of the filter-digester bed over time after fecal matter was deposited.

To compare the overall bacterial community structure of the filter-digester bed over the time series, we used two different beta diversity metrics: Bray–Curtis dissimilarity, which accounts for abundance of OTUs but not their phylogenetic relatedness, and weighted

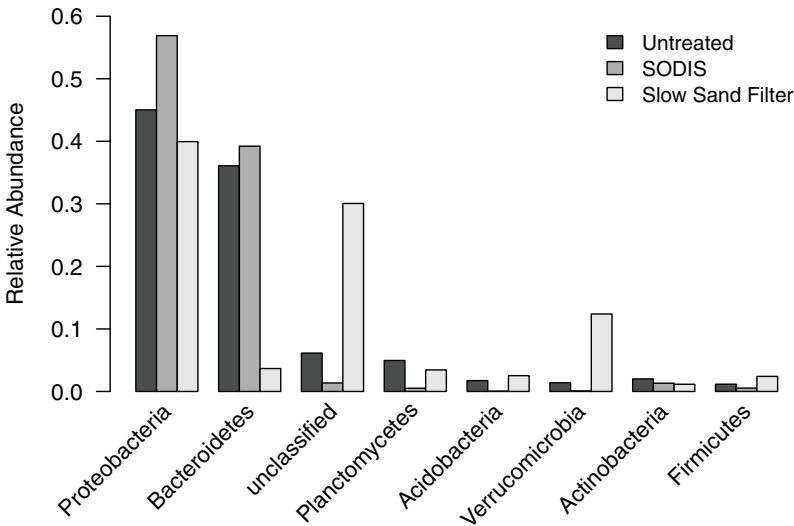

**Figure 4** Relative abundance of phylum-level groups occurring at >2% in at least one of the liquid effluent samples.

UniFrac distance, which assesses phylogenetic relatedness of OTUs and weights branches based on OTU abundance. Dendrograms of the resulting distance matrices showed that samples from the last three time points cluster together (Fig. S3). The Bray–Curtis distance matrix clustered the day 4 sample with the time zero sample, albeit with a large distance between these two samples, whereas the weighted UniFrac distance matrix clustered the day 4 sample with the last three samples. Overall, beta diversity analysis provides additional evidence of a major shift in the bacterial community composition of the filter-digester bed occurring within a few days after deposition of fecal matter.

## Bacterial taxa detected in untreated liquid effluent are more abundant in SODIS-treated effluent but greatly reduced or absent from SSF-treated effluent

To characterize the bacterial community of the liquid effluent from the test system, we collected a sample of effluent on day 15 from a drainage pipe connected to the filtrate chamber at the base of the test system. A portion of this sample was analyzed as "untreated" liquid effluent. We used the remainder to test two different sanitization methods: SODIS and SSF. A sample of the output from each method was analyzed as "treated" liquid effluent. Each of the effluent samples was subsampled to 106,154 sequences.

Analysis of the liquid effluent samples clustered the subsampled sequences into 2,903 OTUs, which were classified into 24 phyla and one "unclassified" group that included all OTUs that could not be classified at the phylum level. Seven of these phyla and the unclassified group occurred at a relative abundance >2% in at least one of the effluent samples (Fig. 4). Proteobacteria was the dominant phylum in all three samples, comprising 45.0% of sequences in the untreated effluent, 56.9% of sequences in the SODIS-treated effluent, and 39.9% of sequences in the SSF-treated effluent. Bacteroidetes was abundant in the untreated and SODIS-treated effluent (36.1% and 39.2% of sequences, respectively),

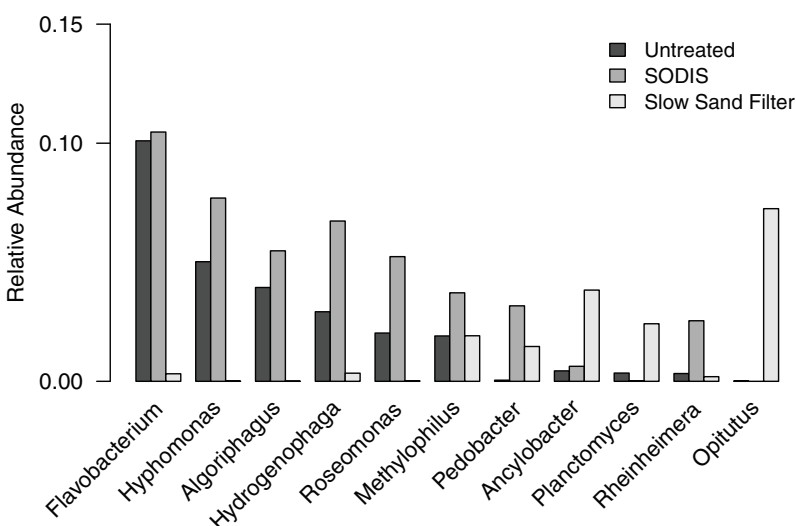

**Figure 5** Relative abundance of genera occurring at >2% in at least one of the liquid effluent samples.

but this phylum comprised only 3.7% of sequences in the SSF-treated effluent. By contrast, the unclassified group and Verrucomicrobia occurred at a higher relative abundance in the SSF-treated effluent compared to the untreated and SODIS-treated effluent. The other four phyla occurred at a relative abundance ≤5% in all liquid effluent samples.

To further examine the differences in bacterial community composition between untreated and treated liquid effluent, we analyzed the OTUs at the level of genus. A total of 11 genera and an "unclassified" group occurred at a relative abundance >2% in at least one of the samples. The untreated effluent and both treated effluents were dominated by the unclassified group, which comprised 48.9–76.0% of sequences and is further discussed below. Five genera (*Flavobacterium*, *Hyphomonas*, *Algoriphagus*, *Hydrogenophaga*, and *Roseomonas*) that occurred at a relative abundance >2% in the untreated effluent were detected at a higher relative abundance in the SODIS-treated effluent (Fig. 5). By contrast, these genera were not prominent members of the SSF-treated effluent community, occurring at a relative abundance <0.4%. In addition, two minor members of the untreated effluent community (*Methylophilus* and *Rheinheimera*) increased in abundance in the SODIS-treated effluent. The abundance of *Methylophilus* was unchanged in SSF-treated effluent. We detected only one sequence classified as *Methylophilus* in the negative controls, therefore we can conclude that the persistence of *Methylophilus* is not an artifact of contamination. In the SSF-treated effluent, three genera that occurred at low relative abundance in the untreated and SODIS-treated effluent (*Ancylobacter*, *Planctomyces*, and *Opitutus*) were detected at a higher abundance.

Because many of the sequences in our genus level analysis were clustered into an "unclassified" group, we analyzed these OTUs separately. Relative abundance in the following analysis refers to the abundance of a particular unclassified OTU compared to the total number of sequences clustered into the unclassified group during the genus level analysis. A total of 14 unclassified OTUs occurred at a relative abundance >2% in at least

one of the three effluent samples (Fig. S4). These OTUs accounted for 61.7% of total sequences in the unclassified group for untreated effluent, 89.7% for SODIS-treated effluent, and 43.5% for SSF-treated effluent.

Of the seven unclassified OTUs that occurred at a relative abundance >2% in the untreated effluent, four of these increased in abundance in the SODIS-treated effluent, paralleling the trend observed at the genus level analysis. In addition, two OTUs that occurred at a relative abundance 1–2% in the untreated effluent also increased in abundance in the SODIS-treated effluent. All six of these OTUs were minor members of the SSF-treated effluent, occurring at a relative abundance <0.08%. The SSF treatment appeared to introduce OTUs into the effluent. Five OTUs that occurred in the SSF-treated effluent were either not detected in the untreated and SODIS-treated effluent or detected only at a low relative abundance (0.03–0.07%). The abundance of one OTU (Otu0012) was largely consistent across the liquid effluent samples. We detected only eight sequences classified into this OTU in the negative controls, therefore we can conclude that the persistence of this OTU is not an artifact of contamination.

Comparing the bacterial community composition of the untreated liquid effluent to that of the filter-digester bed sample also collected on day 15, we found only two genera (*Flavobacterium* and *Algoriphagus*) at a relative abundance >2% in both the liquid effluent and filter-digester bed samples. This suggests that the bacterial community of the liquid effluent from the test system is different from that of the solid filter-digester bed.

## DISCUSSION

Based on analysis of OTUs at different taxonomic levels and beta diversity metrics, a major shift in the bacterial community composition of the filter-digester bed at the site of fecal matter deposition occurs within a few days. The phylum-level profile of the time zero sample is consistent with that of the human gut microbiome (*Hugon et al., 2015*; *Turnbaugh et al., 2007*), which is expected because this sample was collected immediately after depositing human fecal matter. At the genus level, the bacterial community of the time zero sample is dominated by known human gut microbes such as *Bacteroides*, *Blautia*, and *Faecalibacterium* (*Franzosa et al., 2014*; *Rajilić-Stojanović & De Vos, 2014*). With the exception of *Clostridium* XI, these gut-associated genera are detected at extremely low relative abundance (<0.02%) in the filter-digester bed at day 30, and most are already greatly reduced by day 4. By contrast, *Clostridium* XI was detected in the time zero sample and all subsequent samples. This group includes the gastrointestinal pathogen *Clostridium difficile* and other *Clostridium* species, which are anaerobic spore-forming bacteria (*Pérez-Cobas et al., 2014*).

The ability to withstand harsh environmental conditions over extended periods of time by forming spores likely explains why *Clostridium* XI does not disappear from the filter-digester bed by day 30. Persistence of spore-forming *Clostridium* species has been observed in other studies of composting toilets (*Tønner-Klank et al., 2007*), and these species are resistant to chemical treatment of human waste with urea (*Vinnerås et al., 2003*). Treatment of human waste with peracetic acid successfully eliminated spore-forming *Clostridium* species (*Vinnerås et al., 2003*); however, application of this

chemical may not be practical or desirable for communities using the GSAP Microflush toilet. Not all spore-forming bacteria are associated with disease, therefore detection of *Clostridium* XI does not necessarily indicate a threat to public health from compost material, but this result highlights the importance of considering persistence of certain species via spore formation in the filter-digester bed.

In the subsequent samples collected beginning on day 4 from the site of initial fecal matter deposition, taxa were detected that did not appear in the time zero sample, including three new phyla and six new genera. These taxa persisted through to the day 30 sample. The six genera are primarily associated with environmental sources rather than the human gut microbiome. For example, *Flavobacterium* and *Algoriphagus* belong to Bacteroidetes and have been isolated from soil and aquatic environments (*Nedashkovskaya et al., 2007*; *Thomas et al., 2011*). *Pseudoxanthomonas* and *Rhodobacter* belong to Proteobacteria, which increased in abundance over the time series. *Pseudoxanthomonas* has been found as a dominant taxon in maize stover composted by a process using the earthworm *Eisenia fetida* (*Chen et al., 2015*), which are present in the filter-digester bed studied here. These taxa detected beginning on day 4 were likely already established in the filter-digester bed as a result of ongoing maintenance by periodic addition of vegetable scraps.

The data indicate that the composition of the filter-digester bed at the site of deposition of fecal matter shifts from a bacterial community dominated by taxa associated with the human gut to taxa associated with the filter-digester bed itself. The conditions of the filter-digester bed may not be amenable to gut-associated taxa, since many gut bacteria are anaerobes, and the GSAP Microflush toilet is designed to use an aerobic composting process. In addition, bacterial taxa that have become established within the filter-digester bed may outcompete gut-associated taxa that are periodically introduced. In both the test system studied here and the field system deployed in a community setting, the exact composition of the bacterial community of the filter-digester bed will be influenced by how the GSAP Microflush toilet system is maintained and used; for example, how often and what kind of non-fecal material is added to the filter-digester bed. Using the test system as an indicator of the performance of the field system, we have shown how the GSAP Microflush toilet can be used to immediately contain and over time eliminate gut-associated bacteria, thereby improving sanitation and contributing to better public health in rural and peri-urban communities.

Typically, liquid effluent from the GSAP Microflush toilet exits the system into a soak hole that acts as a leach field. However, this design is not suitable for communities with high water tables due to concerns about contamination of water sources. In these situations, alternative sanitization strategies are needed. Here, we compared the effects of SODIS and SSF on liquid effluent from the test system. Bacterial genera that were abundant in the untreated liquid effluent on day 15 are generally associated with environmental sources such as mud and freshwater. *Flavobacterium* and *Algoriphagus* were detected in both the filter-digester bed sample collected on day 15 and the untreated liquid effluent collected on the same day, suggesting that bacteria in the filter-digester bed may enter the filtrate chamber via liquid flowing through the screens and mesh

separating these two compartments. The untreated liquid effluent also included *Hyphomonas*, which is typically associated with marine environments. Its presence may be related to the test system's location on the coast of Narragansett Bay.

The genera and unclassified OTUs that were abundant in the untreated liquid effluent were greatly reduced or not detected in the SSF-treated effluent; however, these taxa increased in abundance in the SODIS-treated effluent. A previous study of SODIS for application with the GSAP Microflush toilet system showed significant reduction of viable *Escherichia coli* over 8 h of exposure to sunlight (*Mecca, Pellock & Bretz, 2013b*). Given this and other studies showing the capacity of SODIS to sanitize contaminated water (*Boyle et al., 2008*; *Caslake et al., 2004*), it is possible that bacterial cells in the SODIS-treated effluent are no longer viable, but DNA is recoverable and detected using the 16S rRNA gene sequencing methods applied here.

In contrast to SODIS, the physical filtration approach of SSF treatment resulted in extremely low or undetectable levels of most genera and unclassified OTUs found in the untreated liquid effluent. This is expected, because this sanitization method is designed to physically remove bacterial cells from the liquid effluent. We noted that SSF-treated effluent included taxa that were not detected in either the untreated effluent or the SODIS-treated effluent. These taxa are likely introduced from the schmutzdecke, which is a naturally formed biological layer of microbes and invertebrates that is the key component of SSF treatment.

## CONCLUSIONS

Overall, analysis of both the filter-digester bed and the liquid effluent of the GSAP Microflush toilet test system illustrated the capacity of the design to isolate and eliminate bacteria associated with human waste, thereby improving sanitation, contributing to the maintenance of safe water sources and protecting public health. The invertebrate-enhanced aerobic composting process reduces fecal-associated taxa in the filter-digester bed within a few days, with the exception of spore-forming *Clostridium* species that are detected at low abundance even after 30 days. In addition, the data presented here provide further insight towards the design of alternative strategies for sanitization of liquid effluent in communities with high water tables.

## ACKNOWLEDGEMENTS

We thank Janet Atoyan at the University of Rhode Island Genomics and Sequencing Center for assistance with 16S rRNA amplicon sequencing. We thank Kyle Edmonds and Lauren Precopio for valuable contributions to establishing protocols for sample processing and data analysis. We are grateful to Jeff Hollister, Pat Schloss and Nicole Sukdeo for advice regarding data analysis.

### Funding

This research was supported by funds from Providence College, including a grant from the Committee on Aid to Faculty Research. This material is based upon work conducted at a

Rhode Island NSF EPSCoR research facility, the Genomics and Sequencing Center, supported in part by the National Science Foundation EPSCoR Cooperative Agreement #EPS-1004057. The funders had no role in study design, data collection and analysis, decision to publish, or preparation of the manuscript.

### Grant Disclosures

The following grant information was disclosed by the authors:
Providence College, including a grant from the Committee on Aid to Faculty Research.
Rhode Island NSF EPSCoR research facility.
Genomics and Sequencing Center.
National Science Foundation EPSCoR Cooperative Agreement #EPS-1004057.

### Competing Interests

The authors declare that they have no competing interests.

### Author Contributions

- Laura E. Williams conceived and designed the experiments, performed the experiments, analyzed the data, contributed reagents/materials/analysis tools, prepared figures and/or tables, authored or reviewed drafts of the paper, approved the final draft.
- Claire E. Kleinschmidt performed the experiments, analyzed the data, authored or reviewed drafts of the paper, approved the final draft.
- Stephen Mecca conceived and designed the experiments, performed the experiments, contributed reagents/materials/analysis tools, authored or reviewed drafts of the paper, approved the final draft.

### Data Availability

Williams, Laura (2018): Liquid effluent sample analysis. figshare. Fileset.
Williams, Laura (2018): Digester sample analysis. figshare. Fileset.
Williams, Laura (2018): Negative control analysis. figshare. Fileset.
Williams, Laura (2018): Mock community analysis. figshare. Fileset.
Williams, Laura (2018): MiSeq data and sample guide. figshare. Fileset.
https://figshare.com/projects/Data_and_R_code_for_Microflush_Composting_Toilet_Microbiome_Project/35342

### Supplemental Information

Supplemental information for this article can be found online at http://dx.doi.org/10.7717/peerj.6077#supplemental-information.

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
