# Peer review of "Bacterial communities in the digester bed and liquid effluent of a microflush composting toilet system"

_PeerJ, doi:10.7717/peerj.6077_

## Round 0.1 · original submission · Minor Revisions

The manuscript was well written and will be acceptable for publication with careful consideration of the reviewers' comments. Each of the points brought up are valid and should be addressed in your revised manuscript. In addition, I would like to see mention of what approach might be used to combat spore forming bacteria you observed at Day 30 (Clostridium XI). Also, I believe the persistence of this gut bacteria should be included within the conclusion. It was very good to see a new approach to evaluating the effectiveness of this type of system through examination of bacterial community populations. I look forward to reading your revised manuscript.

·

Basic reporting

clear and unambiguous

professional structure, figs, tables, raw data shared

Experimental design

original primary research

rigorous investigation performed to high technical & ethical standards

Validity of the findings

Conclusions are well stated.
Data is robust

Additional comments

This is a solid research paper and the conclusions make sense.

I think the paper deserves publication at this level of detail.

Open question for me is the question of at what point the spore forming bacteria of the Clostridium XI which are potentially associated with disease and threats to public health, are they no longer detected.

The safety of the microflush composting system would be further verified if we knew the answer to this question. But it might require months or even a year or more of sampling. I would like to see that work done as a follow-on research paper.

However, this work is worthy of publication and is well done. It is original and important and worth publication

Reviewer 2 ·

Basic reporting

The manuscript is well written with clear structure and use of English language. The manuscript touches upon a lesser known and relevant aspect of such systems. The microbial community is both an indicator of the function of the system as well as its efficiency to remove pathogens. However several aspects in building a case for the study needs to be better emphasized.
(1) Please update the statistics and reference for the line 44 to 49. It is usually better to use the JMP reports since these are the most up-to-date and is the source of all the information used across the UN publications. There is a 2017 progress report.
(2) The system that is under study was designed for rural areas but no information was given as to why it focusses on rural areas. Is it because of the space requirement? Is because they are behind urban sanitation in using safely managed sanitation? Please provide a case on why it is necessary.
(3) The paragraph starting at Line 77 seems to be pointing to studies done within the group on this particular system? This is not immediately clear. Please emphasize this other wise you need to update the literature as there are more recent studies on e.g. vermicomposting
(4) There is no clear hypothesis and no mention of expected microbial community and their importance to the system or expected bacteria based on literature.

Experimental design

While much emphasis has been places on microbial community analysis which has been clearly described, less is given on the test system and parameters of interest. Thus, the study can be significantly improved if the following is addressed:
1. As stated prior, there is no clear hypothesis, this can help to understand the methodology and findings e.g. why sample liquid for effluent treatment at day 15 and not earlier or later?
2. This may also be clearer if a description or process diagram or even a picture is added to show the design of the ‘pilot scale’ system that is used to generate the results. It is unclear what are the operational parameters used to generate the final treated products and this may have influences on the microbial community.
3. Please report the physico-chemical properties of each sample taken e.g. pH, temp, ammonia, VSS, conductivity etc.

Validity of the findings

The data are well reported. However the lack is description of the pilot set up and operation as well as the properties of the samples prevented the authors from identifying areas for further work as well as provide inferences. Some speculations were provided but no information was used to support the speculations for example, no information was provided on the microbial community that was in the conditioned system prior to this experiment. Further, as far as I can tell, the experiments were done only once without repetition. If these were included, the study could potentially be improved. Specific comments includes:
1. Please clearly name the samples take at day 0 for the ‘liquid’ portion or effluent and the sample taken at day 15 for the liquid treatment processes. It is a bit confusing as to which samples you are talking about in 376-380 and 319-321.
2. Line 404 change ‘the’ to ‘to’.

---

## Round 0.2 · accepted · Accept

Thank you for addressing all of the concerns of the reviewers in a careful and conscientious manner. The revised manuscript is improved and acceptable for publication. I look forward to seeing your future studies on the GSAP Microflush toilet system.

#